# Comparative test-retest variability of outcome parameters derived from brain [18F]FDG PET studies in non-human primates

Sébastien Goutal[1], Nicolas Tournier[2☯], Martine Guillermier[1☯], Nadja Van Camp[1], Olivier Barret[1], Mylène Gaudin[1], Michel Bottlaender[2,3], Philippe Hantraye[1], Sonia Lavisse[1]*

**1** Laboratoire des Maladies Neurodégénératives, CEA, CNRS, MIRCen, Université Paris-Saclay, Fontenay-aux-Roses, France, **2** Laboratoire d'Imagerie Biomédicale Multimodale (BioMaps), CEA, CNRS, Inserm, Service Hospitalier Frédéric Joliot, Université Paris-Saclay, Orsay, France, **3** UNIACT, Neurospin, CEA, Université Paris-Saclay, Gif-sur-Yvette, France

☯ These authors contributed equally to this work.
* sonia.lavisse@cea.fr

**Data Availability Statement:** All relevant data are within the paper and its Supporting Information files.

## Abstract

### Introduction

Knowledge of the repeatability of quantitative parameters derived from [18F]FDG PET images is essential to define the group size and allow correct interpretation. Here we tested repeatability and accuracy of different [18F]FDG absolute and relative quantification parameters in a standardized preclinical setup in nonhuman primates (NHP).

### Material and methods

Repeated brain [18F]FDG scans were performed in 6 healthy NHP under controlled experimental factors likely to account for variability. Regional cerebral metabolic rate of glucose (CMRglu) was calculated using a Patlak plot with blood input function Semi-quantitative approaches measuring standard uptake values (SUV, SUV×glycemia and SUVR (SUV Ratio) using the pons or cerebellum as a reference region) were considered. Test-retest variability of all quantification parameters were compared in different brain regions in terms of absolute variability and intra-and-inter-subject variabilities. In an independent [18F]FDG PET experiment, robustness of these parameters was evaluated in 4 naive NHP.

### Results

Experimental conditions (injected dose, body weight, animal temperature) were the same at both imaging sessions (p >0.4). No significant difference in the [18F]FDG quantification parameters was found between test and retest sessions. Absolute variability of CMRglu, SUV, SUV×glycemia and normalized SUV ranged from 25 to 43%, 16 to 21%, 23 to 28%, and 7 to 14%, respectively. Intra-subject variability largely explained the absolute variability of all quantitative parameters. They were all significantly correlated to each other and they were all robust. Arterial and venous glycemia were highly correlated (r = 0.9691; *p*<0.0001).

**Funding:** This work was partially funded by ANR-11-INBS-0011 - NeurATRIS: A Translational Research Infrastructure for Biotherapies in Neurosciences. There was no additional external funding received for this study.

**Competing interests:** The authors have declared that no competing interests exist.

## Conclusion

[$^{18}$F]FDG test-retest studies in NHP protocols need to be conducted under well-standardized experimental conditions to assess and select the most reliable and reproducible quantification approach. Furthermore, the choice of the quantification parameter has to account for the transversal or follow-up study design. If pons and cerebellum regions are not affected, non-invasive SUVR is the most favorable approach for both designs.

## Introduction

Positron emission tomography (PET) imaging using 2-deoxy-2-[$^{18}$F]fluoro-D-glucose (FDG) remains so far the most accurate *in vivo* method for the investigation of regional glucose metabolism in the living brain. In the brain, glucose metabolism is tightly linked to neuronal and synaptic functions. [$^{18}$F]FDG PET therefore provides a convenient tool to non-invasively monitor changes in neuronal activity in pathophysiological states. [$^{18}$F]FDG PET is a translational imaging method, which can be performed either in humans or in animals going from rodents to large animal species such as nonhuman primates (NHP) [1]. Though invasive in rodents, quantitative [$^{18}$F]FDG PET is only minimally invasive in NHPs, which facilitates its use as a close-human animal model. The high-resolution of preclinical PET scanners in particular allows determination of the level of glucose metabolism in discrete brain regions. Over the last four decades, [$^{18}$F]FDG brain PET has mostly been used in the context of neurodegenerative diseases and brain tumor research [2]. As non-invasive technique, [$^{18}$F]FDG PET has also been used to monitor the regional effects of potential CNS drugs and/or pharmacological challenges with repeated scans [3–6]. More recently, [$^{18}$F]FDG brain PET was also applied to unravel brain networks in healthy and diseased brains [7]. Many quantification approaches exist to determine brain glucose metabolism, hence questioning the accuracy of the different [$^{18}$F]FDG PET quantification methods.

In the nuclear medicine practice, [$^{18}$F]FDG PET images are often analyzed qualitatively, using visual comparison of static images to identify regions with abnormal glucose uptake compared with normal surrounding brain tissue. PET data can also be interpreted from the time-activity curves (TAC) in selected brain regions which are usually reported as percentage of injected dose (%ID/mL) or standardized uptake value (SUV) *versus* time. SUV (g/mL) is defined as the tissue concentration (MBq/mL) divided by injected activity per body weight or lean body mass (MBq/g) [8] and represents a normalization approach reflecting the degree of metabolic activity in healthy or pathological tissues.

Peripheral glucose metabolism drastically impacts the plasma concentration of FDG and its brain kinetics, hence influencing the estimation of brain glucose metabolism. Many authors have discussed the multiple factors that affect the SUV index, such as kinetics of tracer uptake in the brain [9]. Methods exist to approach absolute quantification of brain glucose utilization or, at least, reduce the effect of unintended and non-specific variability of brain [$^{18}$F]FDG PET data between conditions. Estimation of the cerebral metabolic rate of glucose (CMRglu) can be achieved using full kinetic modelling of [$^{18}$F]FDG PET data using the irreversible two-compartment model or the graphical method of Patlak [10]. The outcome parameter integrates glycemia and [$^{18}$F]FDG plasma kinetics (i.e the arterial input function), and is technically demanding as it requires arterial blood sampling at multiple time-points. This is invasive, uncomfortable and can even be sometimes considered unethical for the patient. Further simplification or normalization methods for [$^{18}$F]FDG PET have therefore been proposed to

overcome this limitation and avoid arterial catheterization in patients [11–13]. However, many factors of variability still exist and full kinetic modelling remains the gold-standard method for brain [18F]FDG PET quantification, notably in preclinical settings.

Knowledge of the repeatability of estimated outcome parameters derived from [18F]FDG PET is particularly relevant for longitudinal follow-up studies assessing brain disease progression and/or therapeutic efficacy. The determination of group sizes is essential to [18F]FDG PET studies design for correct interpretation of changes measured during follow-up. Also, it is essential to control for experimental factors so as to minimize additional sources of variability on data measurement, analysis or interpretation and their respective impact on overall test-retest variability during repeated brain [18F]FDG PET experiments.

Standardization of [18F]FDG PET imaging has previously been discussed in the clinical context [14]. Preclinical test-retest studies of cerebral [18F]FDG PET (with arterial blood sampling) have been performed by various groups but in rodents only [15, 16]. To the best of our knowledge, no preclinical study in NHP has yet reported the evaluation of the repeatability and accuracy of the different [18F]FDG PET quantification approaches in a standardized preclinical setup. In the present study, we performed repeated brain [18F]FDG PET acquisition in six healthy NHP using a standardized procedure to control for experimental factors. First, the test-retest variability of PET data metabolism outcome parameters was assessed, and compared for accurate interpretation of brain [18F]FDG images. Finally, we evaluated the robustness of the outcome parameters by considering their respective confidence interval on an independent group of naive NHP.

## Material and methods

### Animals and housing

All primate experiments performed in this study were approved by the local ethics committee Comité d'Ethique en Experimentation Animale, CETEA n˚44 (authorization #12_074) and the Ministry of Higher Education, Research and Innovation (MESRI) ((APAFIS#389) 2015032716213569.02.), were conducted according to European regulations (EU Directive 2010/63) and in compliance with Standards for Humane Care and Use of Laboratory Animals of the Office of Laboratory Animal Welfare (OLAW–n˚#A5826-01) in a facility authorized by local authorities (authorization n˚#B92-032-02).

Animals were originally obtained from approved breeding facilities and vendors (Noveprim; Ferney S.E.; Mahebourg Mauritius Island). The main study used six adult male cynomolgus macaques (Macaca fascicularis) of about 4 years of age (4.2 ± 0.8 yrs) and 5.8 ± 1.1 kg average weight. The robustness study included four adult cynomolgus male primates of about 4 years of age (4.2 ± 0.7 yrs) and 4.5 ± 0.9 kg average weight. Animals were housed in pairs or social groups depending on compatibility in a caging system with multiple compartments that were cleaned at least once every two weeks while sawdust was changed twice a week.

Animals were housed under a 12-hour light-dark cycle, temperature was maintained at 22 ±1˚C and humidity at 50% in the animal quarters. Animals were fed with a commercial monkey chow, supplemented daily with fruits and vegetables changed on a daily basis as part of the enrichment program and drinking water was changed daily and was available ad libitum. Throughout the study, animals were checked daily by the technicians to evaluate their physical and clinical condition, food and water consumption. Environmental enrichment activities included grooming contact, perches, opaque and colored cage panels, and access to additional environment enrichment devices such as mirrors, swings, foraging devices, and manipulation and feeding toys. All imaging procedures were performed under ketamine/xylazine anesthesic induction (10 and 1mg/kg, respectively, intramuscular injection) and animals were then

maintained anesthetized using an intravenous (i.v.) infusion of propofol (1mL/kg/h) to minimize pain. Animals were not euthanized at the end of these experiments. None of the animals became severely ill during the course of the study or none required euthanasia prior to their experimental endpoint.

## Magnetic resonance imaging

Magnetic resonance imaging (MRI) acquisition procedure was previously described [1]. Briefly, MRI was performed on each NHP in order to delineate cerebral regions on co-registered PET images. They were placed in the magnet in a sphinx position with the head fixed in a stereotactic MRI-compatible frame (M2E, France) and heated by a hot air flux with their temperature and respiration parameters monitored remotely.

Acquisitions were performed on a horizontal 7T Agilent scanner (Palo Alto, CA, USA) equipped with a gradient coil reaching 100mT/m (300μs rise time). A surface coil (RAPID Biomedical GmbH, Rimpar, Germany) was used for transmission and reception. T2-weighted images were acquired using a high-resolution 2D fast spin-echo sequence (469×469 μm$^2$ in-plane resolution, 1 mm slice thickness, 70 slices), with echo time TE/ Repetition time TR = 20/8000 ms, 5 echoes, effective TE = 52.5 ms and acquisition time Tacq = 43 min.

## PET imaging

**PET acquisition.** [18F]FDG PET imaging was performed in all NHPs after overnight fasting with free access to water. Animals were initially sedated with the ketamine/xylazine mixture and anesthetized under propofol starting with an i.v. bolus followed by infusion in a saphenous vein. The other saphenous vein was catheterized for venous sampling. The femoral artery opposite to the tracer/propofol injection site was cannulated for arterial blood sampling. Primates were wrapped in a heat blanket to maintain body temperature throughout the protocol. Blood pressure, heart rate, respiration rate and oxygen saturation were remotely monitored and registered continuously before and during the acquisition.

Animals were placed in a PET dedicated stereotactic-like animal holder to immobilize the head by ear bars. PET imaging was performed on a Concorde Focus220 microPET scanner providing high sensitivity and high spatial resolution (FHWM = 2 mm) (Siemens, TN, USA). A 17-min-transmission scan (using a $^{57}$Co point source) was performed prior to PET acquisition to allow for attenuation correction, followed by a 60 min dynamic emission scan acquired in 2D mode. Data acquisition started simultaneously with the i.v. bolus injection of [18]FDG (into the same vein as for propofol injection) using an automated injection pump at a rate of 4 mL/min during 30 seconds. PET images were corrected for radioactive decay, scatter, attenuation and detector inhomogeneity, were rebinned in 27 frames (4×15, 4×30, 2×60, 5×120, 3×300, 9×600 sec.) and reconstructed as previously described (OSEM-2D) [17].

To estimate the test-retest variability, a second scan was performed under exact same conditions after an interval of 27 ± 6 days. The time interval was selected to reduce stress induced by close, repeated food deprivation, anesthesia and animal handling. Overall average of [18F]FDG injected dose (ID) was of 169.2 ± 9.0 MBq.

Arterial blood samples were manually drawn from the femoral artery just before and during each PET acquisition to establish the kinetics in arterial plasma of [18F]FDG (arterial input function, AIF) and glucose concentration. After [18F]FDG injection, 2 ml of arterial blood was sampled every 15 s in the first 2 min and then at 2.5, 3, 5, 10, 20, 40 and 60 min after the injection. Blood samples were centrifuged and 0.2 mL plasma samples were counted in a gamma counter (Wizard$^2$, PerkinElmer) that was cross-calibrated with the PET scanner. Area under the curve of AIF was calculated from 0 to 60 minutes post injection.

Arterial glycemia (Gly) and venous blood samples were withdrawn simultaneously just before [18F]FDG injection. Both measurements were compared at test and retest sessions. All glycemia samples were measured in duplicate with a GM9 glucose analyser (Analox, Stourbridge, UK), which was calibrated at 5 mmol/L using standard solutions. The bias and repeatability of the GM9 analyzer was determined from 10 measures of a glucose standard.

The four additional and independent animals were scanned once under the same conditions so as to address the robustness of the quantification methods.

**PET image analysis.** Time frames were summed to create an integrated image for automatic co-registration with the T2-weighted MRI using a dedicated PMOD software (PMOD Technologies, version 3.6, Zurich, Switzerland). Anatomical regions of interest (ROI) were automatically delineated on individual MRI images using the CIVM atlas from the Primatologist segmentation pipeline for the macaque brain [18] and were as follows: thalamus, globus pallidus, caudate, putamen, global cortex, midbrain, pons, cerebellum, frontal and occipital cortices. Kinetics of the radioactivity of each ROI was then extracted from the dynamic PET data as a time-activity curves (TACs). For each animal, the same MRI segmentation was applied to extract TACs from test and retest PET images.

The Patlak graphical analysis was applied to the regional TACs in PMOD to quantitatively estimate the cerebral metabolic rate of glucose (CMRglu) under the hypothesis that the tissue includes a compartment in which the tracer is irreversibly entrapped [10, 19]. Linearization started at 30 min and the lumped constant representing the ratio of FDG utilization to actual glucose utilization in the brain was set to 0.34 [20, 21].

In each ROI, mean SUVs (g/mL) were calculated from the plateau of [18F]FDG brain kinetics, between 30 and 60 min post-injection. Additional SUV values were calculated accounting for glycemia (SUVxGly; mmol/L) [22]. Normalization to a "reference" brain region was performed using SUV uptake value in the pons (SUVR$_{pons}$), or in the cerebellum (SUVR$_{cerebellum}$) as already reported [23, 24].

In total, five quantification parameters were computed and used for statistical analyses: CMRglu, SUV, SUVxGly, SUVR$_{pons}$ and SUVR$_{cerebellum}$.

## Statistical analyses

Data are presented as mean ± SD unless otherwise indicated. Statistical analysis was performed using the R project free software version 3.3.1 (*https://cran.r-project.org/*).

The homogeneity of experimental conditions (body weight, temperature and injected dose at test and retest) were analyzed using Wilcoxon signed-rank test. Plasma exposure (AUC$_{plasma}$) across animals were compared using a one-way analysis of variance (the distribution of residues was normal and homogeneous). Comparison of plasma exposure between test and retest was performed using a paired Wilcoxon rank sum test. The stability of the brain concentration of [18F]FDG was assessed using a two-way ANOVA where macaques and times were defined as explanatory variables (normal and homogeneous distribution of residues), followed by Tukey *post hoc* testing in TACs of all anatomical regions of interest. Comparison of the brain concentration of [18F]FDG was performed using a paired Wilcoxon rank sum test. Test-retest variability was estimated for each brain region and quantification parameter using the absolute variability across animals, defined as the absolute difference between values from the first and the second scan divided by the mean of both values ("AbsVar", × 100 when expressed in %). The total variability was evaluated and explained by the contribution of intra-subject and inter-subject variabilities. Intra-subject variability was expressed 1)- through the coefficient of variation (CV%) and 2)- in relative proportion of the total variability. Inter-subject variability (expressed through the CV% and in relative proportion) was deduced from the total

variability, as proposed by Mermet [25]. Exceptional negative inter-subject variability refered to random fluctuations only and was then set to 0. Finally, a 95% confidence interval (95% CI; absolute and relative value) was built. In 3 (out of 68) cases, lower limits of the CI were negative and were thus arbitrarily set to 0. All correlations were performed using Spearman's rank correlation coefficient.

The sample size ($n$) to estimate the cerebral metabolic rate of glucose consumption with a desired precision level, arbitrary set at 20%, was determined for each quantification parameter:

$$n = \left( \frac{t \times s}{a} \right)^2$$

with $n$ being the sample size, $t$ the coefficient of confidence (= 1.96 for a 5% alpha risk), $a$ the acceptable imprecision of the mean (precision level of 20% of mean value) and $s$, the estimated standard deviation of the population.

## Results

### Homogeneity of experimental conditions

Body weight (5.8 ± 1.4 kg and 6.3 ± 1.3 kg) and injected dose (171.8 ± 9.7 MBq and 166.6 ± 8.7 MBq) were not significantly different between the test and retest scans, respectively ($p = 0.4633$; $p = 0.6905$). During PET acquisition, body temperature remained stable for both test- and the retest scan (37.5 ± 0.4 and 37.3 ± 0.5, respectively, $p = 0.53$).

### Arterial input function

Bolus injection of [18F]FDG resulted in a sharp peak at 0.73 ± 0.07 min followed by a slow distribution phase. Plasma exposures ($AUC_{plasma}$ = 94.1 ± 13.7 SUV.min) were not significantly different accross animals ($p = 0.07$), nor scan sessions ($p = 1$), with mean variability at test and re-test sessions of 14%CV and 15%CV, respectively.

### Glycemia

The standard calibration analysis showed that determination of glucose levels in plasma obtained with GM9 was highly accurate (bias < 1%) and reproducible (CV% < 1%). Glycemia measures from arterial samples were similar between test and retest conditions (4.5 ± 0.7 mmol/L and 5.1 ± 1.5 mmol/L; $p = 0.75$). Glycemia measures from arterial and venous samples at both sessions were highly correlated (4.6 ± 1.3 mmol/L and 4.7 ± 1.4 mmol/L, respectively. Spearman r = 0.9691; $p < 0.0001$, slope = 0.972).

### [18F]FDG PET kinetics and quantification

Visual inspection of the [18F]FDG brain PET images (in Bq/cc and in SUV) revealed qualitative difference between test and retest scans, which was confirmed in the TACs (S1 Fig). This difference appears to be global rather than regional as it was greatly reduced in SUVr images. TACs showed rapid uptake of the tracer in all regions with a maximal uptake 3 minutes after injection and a plateau starting at 30 minutes post-injection, with no significant difference between radioactivity values at 30, 40, 50 and 60 min, whatever brain regions ($p > 0.1$) (S1 Fig).

No significant difference of regional [18F]FDG PET quantitative measures was found between the test and the retest sessions ($p > 0.1$). All quantification approaches resulted in similar regional rank order, measures being the highest in the caudate, putamen and frontal

regions and the lowest in the pons. All regional mean SUVs, SUVR, CMRglu values are reported in Table 1.

**CMRglu estimate.** The statistical values of the regional CMRglu are reported in Table 1A. Overall, absolute variability was high and the range of CMRglu variability across regions was

**Table 1. Mean, confidence interval, total, intra- and inter-subject variabilities of outcome parameters between [18F]FDG test and retest scans.**

| A) | CMRglu (μmol/min/100g) | | | | | B) | SUV (g/mL) | | | | |
|---|---|---|---|---|---|---|---|---|---|---|---|
| | Mean ± sd | 95% CI | Abs var | IntraSV | InterSV | Mean ± sd | 95% CI | Abs var | IntraSV | InterSV |
| | (n = 12) | (relative value) | (%; ) | (%; n = 6) | (%; n = 6) | (n = 12) | (relative value) | (%;) | (%; n = 6) | (%; n = 6) |
| | | | | [EV,%] | [EV,%] | | | | [EV, %] | [EV, %] |
| Thalamus | 20.6±10.1 | [0.0;42.8] | 42.8±28.5 | 45.8 [86] | 18.6 [14] | 2.5±0.5 | [1.4; 3.5] | 18.9±16.4 | 18.5 [93] | 5.2 [7] |
| Globus pallidum | 19.5±9.3 | [0.0;40.0] | 34.2±27.3 | 39.5 [67] | 27.7 [33] | 2.2±0.4 | [1.3; 3.1] | 20.2±15.3 | 18.7 [90] | 6.3 [10] |
| Caudate nucleus | 24.9±10.8 | [1.2;48.6] | 41.8±25.7 | 41.1 [89] | 14.3 [11] | 2.7±0.5 | [1.5; 3.8] | 16.4±16.4 | 17.4 [76] | 9.7 [24] |
| Putamen | 26.2±10.9 | [2.2;50.3] | 42.3±22.0 | 38.1 [82] | 17.8 [18] | 2.8±0.5 | [1.7; 3.8] | 16.1±15.7 | 16.5 [90] | 5.4 [10] |
| Cortex | 19.4±8.2 | [1.5;37.5] | 39.9±24.1 | 38.0 [80] | 18.9 [20] | 2.2±0.3 | [1.5; 2.9] | 16.3±12.5 | 14.8 [94] | 3.8 [6] |
| Midbrain | 20.4±9.5 | [0.0;41.3] | 37.1±29.2 | 43.5 [86] | 17.6 [14] | 2.3±0.4 | [1.4; 3.3] | 18.5±16.7 | 17.9 [100] | 0.0 |
| Pons | 15.8±6.9 | [0.7;30.9] | 25.4±29.2 | 38.1 [75] | 22.0 [25] | 1.9±0.3 | [1.3; 2.6] | 19.0±15.4 | 16.6 [100] | 0.0 |
| Cerebellum | 21.1±9.2 | [0.8;41.5] | 33.2±25.6 | 36.0 [65] | 26.1 [35] | 2.3±0.4 | [1.6; 3.1] | 19.4±10.9 | 15.8 [100] | 0.0 |
| Frontal cortex | 26.3±11.1 | [2.0;50.6] | 40.4±26.8 | 40.1 [90] | 13.1 [10] | 2.8±0.5 | [1.5; 4.0] | 16.3±16.6 | 17.6 [76] | 9.9 [24] |
| Occipital cortex | 17.0±9.0 | [0.0;36.8] | 38.3±29.6 | 46.2 [75] | 26.6 [25] | 2.1±0.4 | [1.2; 3.0] | 20.5±18.9 | 21.4 [100] | 0.0 |
| C) | SUVR_pons (unitless) | | | | | D) | SUVR_Cerebellum (unitless) | | | | |
| | Mean±sd | 95% CI | Abs var | IntraSV | InterSV | Mean±sd | 95% CI | Abs var | IntraSV | InterSV |
| | (n = 12) | (relative value) | (%; ) | (%; n = 6) | (%; n = 6) | (n = 12) | (relative value) | (%;) | (%; n = 6) | (%; n = 6) |
| | | | | [EV,%] | [EV, %] | | | | [EV, %] | [EV, %] |
| Thalamus | 1.3±0.1 | [1.1 ;1.5] | 11.4±8.4 | 10.2 [100] | 0.0 | 1.0±0.1 | [0.8 ;1.2] | 8.7±4.5 | 6.8 [96] | 1.3 [4] |
| Globus pallidum | 1.1±0.1 | [0.9 ;1.3] | 9.5±9.9 | 9.9 [99] | 0.9 [0.8] | 0.9±0.1 | [0.7 ;1.1] | 6.8±4.9 | 5.9 [80] | 3.0 [20] |
| Caudate nucleus | 1.4±0.2 | [1.0 ;1.8] | 13.0±12.8 | 13.3 [100] | 0.0 | 1.1±0.1 | [0.9 ;1.4] | 9.8±7.5 | 8.5 [84] | 3.7 [16] |
| Putamen | 1.4±0.1 | [1.2 ;1.7] | 11.8±11.6 | 11.8 [100] | 0.0 | 1.2±0.1 | [1.0 ;1.4] | 8.6±6.8 | 7.5 [98] | 0.9 [2] |
| Cortex | 1.1±0.1 | [0.9 ;1.4] | 10.3±11.3 | 10.9 [100] | 0.0 | 0.9±0.1 | [0.7 ;1.1] | 7.1±5.9 | 6.3 [100] | 0.0 |
| Midbrain | 1.2±0.1 | [1.0 ;1.4] | 9.1±6.7 | 7.9 [100] | 0.0 | 1.0±0.1 | [0.8 ;1.2] | 6.4±5.0 | 5.7 [92] | 1.6 [8] |
| Pons | | | | | | 0.8±0.1 | [0.6 ;1.0] | 6.8±6.7 | 6.4 [96] | 1.2 [4] |
| Cerebellum | 1.2±0.1 | [1.0 ;1.4] | 6.8±6.7 | 6.5 [99] | 0.3 [0.1] | | | | | |
| Frontal cortex | 1.4±0.2 | [1.0 ;1.8] | 13.9±14.6 | 15.2 [100] | 0.0 | 1.2±0.1 | [1.0 ;1.4] | 10.7±8.2 | 9.5 [78] | 5.1 [22] |
| Occipital cortex | 1.1±0.1 | [0.9 ;1.3] | 10.5±8.8 | 9.7 [100] | 0.0 | 0.9±0.1 | [0.7 ;1.1] | 9.8±7.2 | 8.5 [100] | 0.0 |
| E) | SUV*Gly (mmol/L) | | | | | | | | | | |
| | Mean±sd | 95% CI | Abs var | IntraSV | InterSV | | | | | | |
| | (n = 12) | (relative value) | (%; ) | (%; n = 6) | (%; n = 6) | | | | | | |
| | | | | [EV, %] | [EV, %] | | | | | | |
| Thalamus | 12.2±4.6 | [2.1;22.4] | 27.8±31.4 | 39.4 [100] | 0.0 | | | | | | |
| Globus pallidum | 10.9±4.1 | [1.9;19.8] | 27.3±30.9 | 38.8 [100] | 0.0 | | | | | | |
| Caudate nucleus | 13.3±4.6 | [3.2;23.5] | 27.0±31.0 | 36.8 [100] | 0.0 | | | | | | |
| Putamen | 13.8±4.7 | [3.5;24.2] | 26.2±30.8 | 36.2 [100] | 0.0 | | | | | | |
| Cortex | 10.9±3.5 | [3.2;18.6] | 24.9±29.1 | 33.8 [100] | 0.0 | | | | | | |
| Midbrain | 11.7±4.2 | [2.4;21.0] | 26.9±30.7 | 38.2 [100] | 0.0 | | | | | | |
| Pons | 9.6±3.1 | [2.8;16.3] | 24.1±29.2 | 33.8 [100] | 0.0 | | | | | | |
| Cerebellum | 11.6±3.8 | [3.2;20.0] | 22.8±27.6 | 33.0 [100] | 0.0 | | | | | | |
| Frontal cortex | 13.7±4.6 | [3.5;23.8] | 27.9±30.3 | 36.2 [100] | 0.0 | | | | | | |
| Occipital cortex | 10.5±4.3 | [1.0;19.9] | 26.7±35.1 | 43.0 [100] | 0.0 | | | | | | |

[.] refers to explained variance (EV). Estimates and absolute variability expressed as mean ± SD. InterSV = inter-subject variability; IntraSV = intra-subject variability.

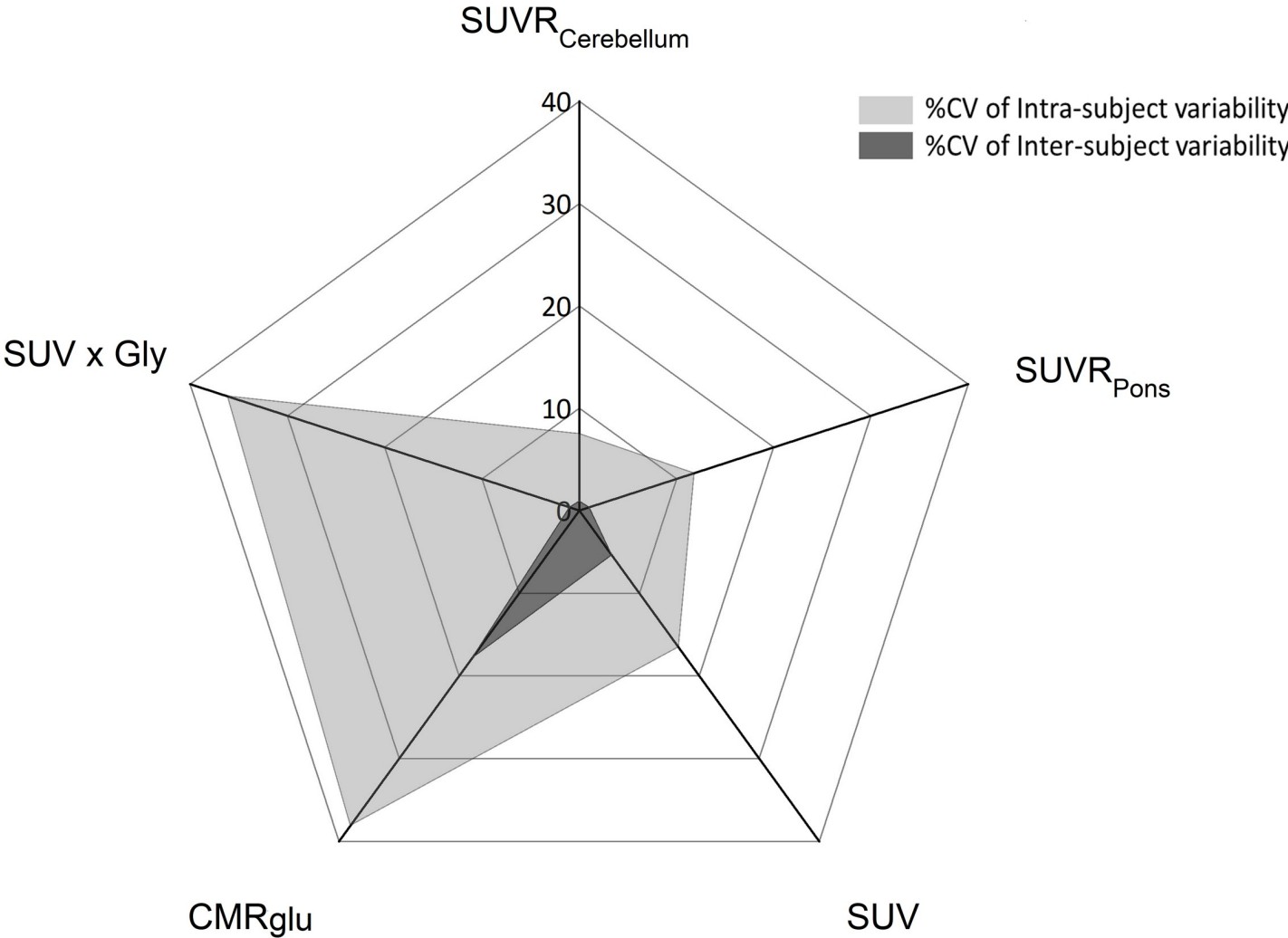

**Fig 1. Mean intra- (n = 6) and inter-variabilities (n = 6) (expressed in CV%) of all [¹⁸F]FDG outcome parameters in the putaminal region.**

from 25.4 to 43%. Lowest variability (25.4%) was found in the pons region. No significant CMRglu difference was reported across regions and intra-subject variability explained between 65 and 90% of the total variability (see Fig 1).

**SUVs and SUVR.** All statistical measures of SUV and SUVR parameters are reported in Table 1B–1E.

*SUV*. Absolute variability of SUV data ranged from 16 to 21% with no regional distinction (global mean = 18.1 ± 1.7 g/mL) and was lower compared to CMRglu absolute variability. The intra-subject CV% (global mean = 17.5 ± 1.8%) was 2.6-fold higher than the inter-subject coefficient of variation in all anatomical ROIs (global mean = 6.7 ± 2.5%) and 2-fold lower than for CMRglu intra-subject variability). In other terms, intra-subject variability of SUV explained between 76 and 100% of the total variability while it explained 65 to 90% of the total variability for CMRglu (Fig 1).

*SUVR*. Normalization of SUV by a reference region decreased the absolute variability down to 7 to 14% for SUVRpons and SUVRcerebellum. Overall variability of SUVR was essentially explained by intra-subject variability ranging from 78 to 100% for SUVRcerebellum and almost 100% for SUVRpons across regions, whereas much lower inter-subject variability was estimated (lower than 5% for SUVRpons and SUVRcerebellum) (Fig 1).

*SUV×Gly*. Correction for arterial glycemia did not improve the absolute variability as compared to SUV: absolute variability ranged between 23 to 28% across brain regions. In all brain regions, intra-subject variability was higher while inter-subject variability was lower in comparison to SUV. Finally, the absolute variations of SUV×Gly (~ 26%) was lower than those obtained for CMRglu (~ 37%) but inter-subject variability was negligible in comparison to CMRGlu.

**Correlations between the quantification parameters.** All quantification parameters were significantly correlated to each other: CMRglu correlated similarly to each of the other [18F] FDG PET measures but correlation was weaker with SUV (r = 0.35). All correlations and statistical significances are provided in Table 2.

## Sample size

Precise estimation (mean±20%) of outcome parameters derived from brain [18F]FDG PET requires sample size of n = 22, n = 5 and n = 6 NHPs for CMRglu, SUV and SUV×Gly, respectively. On the other hand, SUVR had a low variability which allowed to reach this precision with only 2 NHP.

## Robustness of the outcome parameters

The 95% confidence interval was determined on the test-retest dataset for each outcome parameter (Table 1). All SUV×Gly and CMRglu values of the robustness group lied within the confidence interval. In addition, 95% of the regional values of SUV, SUVR$_{pons}$ and SUVR$_{cerebellum}$ were found within the confidence interval (Fig 2).

## Discussion

[18F]FDG has been widely used in clinical and preclinical studies for few decades but this study provides for the first time test-retest variabilities of different quantification parameters in NHP under standardized conditions. Many experimental settings can affect the estimation of glucose consumption in the brain using [18F]FDG PET [22, 26, 27]. In this test-retest study, we aimed at controlling most of these settings to study their impact on various outcome parameters derived from [18F]FDG PET images in NHP. [18F]FDG PET scans were performed on six healthy NHP under the following conditions: experiments were performed in male macaques only [16] with constant and monitored temperature throughout the experiments; the same anesthesia protocol and experimental procedures were applied [28]; all animals were fasted for the same time period before each imaging session [22]; the time interval between PET sessions was constant; the same reconstruction method was applied for all scans [9] and finally, automatic tracer injection and automatic image segmentation were performed to limit

**Table 2. Spearman's rank correlations between outcome parameters from the [18F]FDG test-retest study.**

|  | CMRglu | SUV | SUVR$_{pons}$ | SUVR$_{Cerebellum}$ |
|---|---|---|---|---|
| SUV | 0.35** |  |  |  |
| SUVR$_{pons}$ | 0.66**** | 0.67**** |  |  |
| SUVR$_{Cerebellum}$ | 0.53**** | 0.69**** | 0.91**** |  |
| SUV*Gly | 0.66**** | 0.67**** | 0.68**** | 0.60*** |

****p-value < 0.0001

*** p-value < 0.001 and

**p-value < 0.01 (n = 6 at test and at retest sessions).

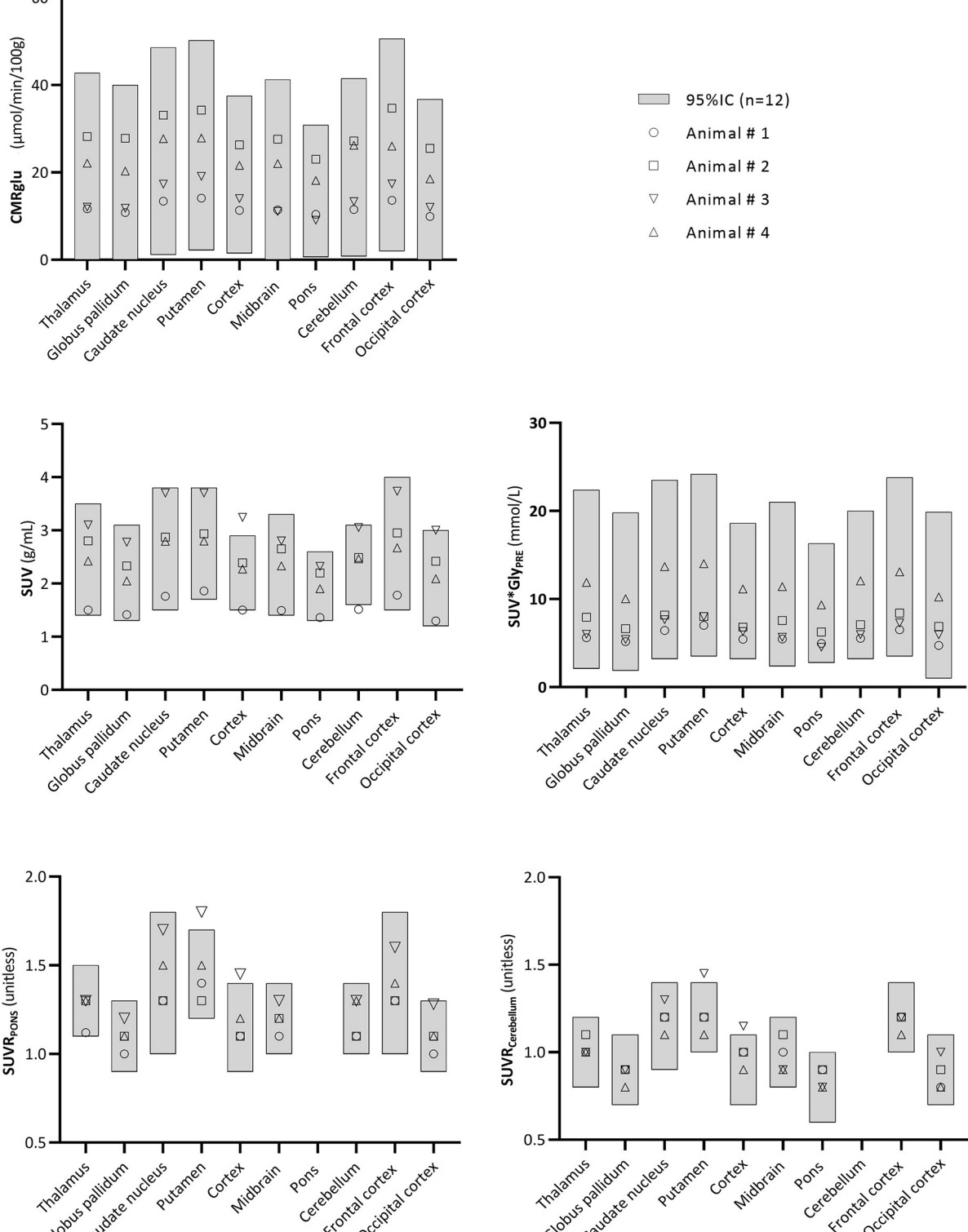

**Fig 2. Robustness study.** Grey boxes represent the 95%IC of each parameter. Four naive animals were considered for this study and regional estimates are represented for each animal and region.

the investigator-induced variability. Under these highly standardized conditions, quantification of outcome parameters were expected to be largely reproducible and indeed no significant differences were observed in any of these parameters between test- and retest PET sessions.

Various groups have reported test-retest [$^{18}$F]FDG variability in preclinical studies but in rodents only. Sijbesma *et al.* [16] have focused on the rat gender difference in cerebral metabolism and reported non negligible absolute variations in different brain regions for CMRglu (mean of 13.4 and 19.6%), SUV (1.61 and 9.12%) and SUV×Gly (9.3 andd 11.9%) in anesthesized male and female rats, respectively. Huang *et al.* estimated the CMRglu parameter and associated variability in two anesthesized rats with 6 repeated [$^{18}$F]FDG PET scans and arterial input function measurements [15]. They concluded that reapeated PET procedure could be feasible in small anesthetized animals when implementing the surgical methods they developed.

In NHPs, arterial catheterization is relatively feasible and repeated blood sampling is well tolerated after a recovery period [29]. Therefore, NHPs are excellent models for full kinetic modeling of radioligands, allowing the comparison of alternative simplified measures with the gold-standard CMRglu calculated by the Patlak method. The CMRglu equation involves the influx rate of [$^{18}$F]FDG ($K_i$), the blood glucose concentration and the lumped constant, accounting for the differences in transport and phosphorylation between glucose and deoxyglucose [30]. Similar glucose levels in male cynomolgus macaques (of the same age as in our study) have been previously reported in few studies: Wu et al. [31] reported a glycemia range of 2 to 4 mmol/L and Guo et al. found glycemia values around 4.7 mmol/L [32]. Brain CMRglu estimates in the present study were also found in the expected and previously reported range [33] but encompassed non negligible variability. Inter and intra-subject variabilities could be partly due to anesthesia injected to the animals. Indeed, the effects of different anesthetic drugs on glucose metabolism in the human brain have been recently reported [34] showing that propofol decreased glucose metabolism compared to saline placebo. In preclinics, studies using isoflurane anesthesia reported a higher glucose level post-scan compared to pre-scan in rats, suggesting a different impact of isoflurane on glycemia [22, 35]. Furthermore, some studies reported that ketamine combined with xylazine induces transient acute hyperglycemia lasting up to 25 minutes in rats [36, 37]. This was confirmed in the rhesus monkey where glucose levels were increased up to 60 minutes after ketamine/dexmedetomidine anesthesia as compared to ketamine only that did not elevate glucose level [38].

Ideally, metabolic brain studies should be performed on unanesthetized awake animals. But routine imaging of a conscious NHP is out of reach and the vast majority of PET studies in NHP are conducted in anesthesized animals. Therefore a standardized sedation protocol requires to maintain stable glycemia and hence reliable measurement of glucose metabolism in primate research. The test-retest procedure would have to be reiterated for each anesthesia condition to determine the intrinsic variability of each quantification parameter and select the most reliable and robust one(s) under experimental standardized conditions and for a given experimental design.

The choice of the processing and quantification methods has also to be considered to explain variability. In our study, the normalized SUV (SUVR) and especially the SUVR$_{cerebellum}$ showed low test-retest variability, indicating a global rather than regional difference between PET sessions. The residual variability might be related in part to our processing pipeline, particularly to the potential PET/MRI misalignment remaining between the two PET sessions of each animal. However, we would expect this local variability to be small compared to other factors affecting more globally the test/retest variabililty of SUV and CMRGlu parameters.

Efforts to decrease the intra- and inter-subject variability in [$^{18}$F]FDG brain PET imaging may help gaining statistical power to detect therapeutic effects or disease progression patterns over time within the same subject panel or between groups with a limited number of subjects.

In our experimental setup, we calculated that the sample size required to accurately determine brain glucose consumption was very different depending on the quantification method applied. Normalization of SUV by a reference region (such as pons or cerebellum) reduced the sample size down to 2 animals because of their much lower inter- and intra-individual variabilities. We noticed here that accounting for glycemia in SUV (SUV×Gly) did not reduce variability, in agreement with previous report [39].

We found that [$^{18}$F]FDG PET outcome measures were significantly inter-correlated across regions and animals. Within-animal correlations (as its own control) were higher than global correlations, favouring individual follow-up experimental design. Nevertheless, significant global correlations (across all animals) between SUV/SUVR and CMRglu parameters underline the possible use of non-invasive procedures instead of arterial sampling to estimate change in brain glucose metabolism over time or treatments. Furthermore, we compared glycemia measurements from arterial and venous blood to avoid arterial collection in animals and found that both measurements were highly correlated. This means that glycemia may be measured in NHP either from venous or arterial blood as long as samples are collected similarly throughout the entire study.

Outcome quantification parameters of [$^{18}$F]FDG images have to be selected upon the design of the imaging protocol. For cross-sectional studies, it is preferable to use a quantification parameter with a low inter-subject variability and for that, SUV, SUV$R_{pons}$ and SUV$R_{cerebellum}$ are excellent candidates. CMRglu showed a reasonable inter-subject variability as well (~20%). On the other hand, SUV$R_{pons}$ and SUV$R_{cerebellum}$ can only be considered in patients that do not suffer from a pathology affecting the cerebellum or the pons region. For longitudinal studies, a low intra-subject variability is needed to accurately follow-up each individual. SUV$R_{pons}$ and SUV$R_{cerebellum}$ fulfill also these criteria, but alternatively SUV can be considered if the cerebellum or the pons are involved in the patient's pathology.

## Conclusion

Altogether, this study suggests that preclinical [$^{18}$F]FDG PET imaging requires highly controlled and standardized experimental conditions to minimize variability and reduce the size of animal cohorts. Definition of standardized protocols enables consideration of preclinical multicentric studies to further reduce animal cohorts in each research site. Appropriate choice of anesthetic drugs and outcome parameters is crucial and should be selected with respect to the study design with the aim to reduce variability and improve the sensitivity and accuracy of [$^{18}$F]FDG PET images to detect changes in glucose brain metabolism.

## Supporting information

**S1 Fig. Representative time activity curves and SUV PET images.** A)- Time activity curves of a representative animal in the pons and thalamus regions; B)- SUV PET images of the same animal at test and retest sessions.
(TIFF)

**S1 File.**
(XLSX)

## Acknowledgments

Authors are thankful to Leopold Eymin and Sophie Lecourtois for animal handling and PET acquisition. They also thank Romina Aron-Badin for animal care support and MRI acquisitions.

## Author Contributions

**Conceptualization:** Nicolas Tournier, Philippe Hantraye, Sonia Lavisse.

**Formal analysis:** Sébastien Goutal, Sonia Lavisse.

**Investigation:** Martine Guillermier, Nadja Van Camp, Mylène Gaudin, Sonia Lavisse.

**Methodology:** Martine Guillermier, Sonia Lavisse.

**Project administration:** Sonia Lavisse.

**Resources:** Philippe Hantraye.

**Supervision:** Nicolas Tournier, Philippe Hantraye, Sonia Lavisse.

**Validation:** Sébastien Goutal, Nadja Van Camp, Olivier Barret, Michel Bottlaender.

**Visualization:** Sébastien Goutal, Nicolas Tournier, Michel Bottlaender, Sonia Lavisse.

**Writing – original draft:** Sébastien Goutal, Nicolas Tournier, Olivier Barret, Michel Bottlaender, Sonia Lavisse.

**Writing – review & editing:** Sébastien Goutal, Nicolas Tournier, Martine Guillermier, Nadja Van Camp, Olivier Barret, Mylène Gaudin, Michel Bottlaender, Philippe Hantraye, Sonia Lavisse.

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
