## [Decision Letter · Decision Letter 0]

10 Aug 2020

PONE-D-20-17186

Comparative test-retest variability of outcome parameters derived from brain [18F]FDG PET studies in non-human primates

PLOS ONE

Dear Dr. Lavisse,

Thank you for submitting your manuscript to PLOS ONE. After careful consideration, we feel that it has merit but does not fully meet PLOS ONE’s publication criteria as it currently stands. Therefore, we invite you to submit a revised version of the manuscript that addresses the points raised during the review process.

We look forward to receiving your revised manuscript.

Kind regards,

Matteo Bauckneht

Academic Editor

PLOS ONE

Journal Requirements:

2. Thank you for including the following funding statement in the acknowledgements sectionn of your manuscript; "This work was partially funded by ANR-11-INBS-0011 - NeurATRIS: A Translational Research Infrastructure for Biotherapies in Neurosciences."

"The authors received no specific funding for this work"

3. 

We note that you have included the phrase “data not shown” in your manuscript. Unfortunately, this does not meet our data sharing requirements. PLOS does not permit references to inaccessible data. We require that authors provide all relevant data within the paper, Supporting Information files, or in an acceptable, public repository. Please add a citation to support this phrase or upload the data that corresponds with these findings to a stable repository (such as Figshare or Dryad) and provide and URLs, DOIs, or accession numbers that may be used to access these data. Or, if the data are not a core part of the research being presented in your study, we ask that you remove the phrase that refers to these data.

Reviewers' comments:

Reviewer's Responses to Questions

**Comments to the Author**

1. Is the manuscript technically sound, and do the data support the conclusions?

Reviewer #1: Yes

Reviewer #2: Yes

Reviewer #3: Yes

2. Has the statistical analysis been performed appropriately and rigorously? 

Reviewer #1: Yes

Reviewer #2: I Don't Know

Reviewer #3: Yes

3. Have the authors made all data underlying the findings in their manuscript fully available?

Reviewer #1: Yes

Reviewer #2: Yes

Reviewer #3: Yes

4. Is the manuscript presented in an intelligible fashion and written in standard English?

Reviewer #1: Yes

Reviewer #2: Yes

Reviewer #3: Yes

5. Review Comments to the Author

Reviewer #1: The authors performed FDG PET brain scans on 6 cynomolgus macaques about one month apart. The parameters include CMRglu, SUVmean, SUVxGlu, SUVRpons and SUVRcerebellum. They also scanned 4 additional animals to see if the values of these parameters fall within the confidence interval defined by the previous 6 animals. The most stable parameter was SUVR with variability between 7-14%, while the variability of CMRglu reached 25-43%. It was concluded SUVR could be used for quantification if pons or crebellum was not affected by the experimental conditions.

This is a well-designed study with clear descriptions of the methodology to allow reproducibility. It provided absolute numbers to compare the variability of various quantification methods across the brain regions. Congratulations to the authors for their deeds and recommendation for publication to share their results.

Reviewer #2: The manuscript presented by Goutal and co-authors explore the repeatability of quantitative parameters from FDG-PET studies in nonhuman primates and also compare them. This is a well-written paper, with a clear and concise text where objectives are well defined, and introduction and discussion supported by a good argumentation. In general, methodology is adequate and experimental methods are described correctly.

However, there are some points that would need to be revised or discussed by the authors in order to improve the quality of the work.

- The authors have carried out a test-retest study controlling all the experimental conditions and have analyzed the reproducibility of the results by the study of intra-subject and inter-subject variability. However, another important measure which can be calculated from test-retest data is the reliability of the measurements by means intraclass correlation coefficient (ICC). Why authors don’t have calculated this coefficient?

- In “PET acquisition” section (page 9, line 130) authors should detail the times at which arterial blood samples were collected to obtain arterial input function.

- For each PET study, authors acquired a 60 min dynamic emission scan (Page 8, line 120). Which temporal sequence did the authors use to reconstruct the studies? Please, explain in “PET acquisition“ section.

- Which statistical test was performed to analyze the homogeneity of experimental conditions (body weight and injected dose)? Please, refer in “Statistical Analyses” section.

- The authors have used a segmentation pipeline to automatically delineate the anatomical regions of interest for each macaque. However, it is usual to spatially normalize brain images to a common stereotaxic space to apply a predetermined template of VOIs. Probably, the method used to quantify the different regions also influences the variability observed in the study. Although the authors discuss the influence of different types of anesthesia on the results of the study (page 19, lines 349-352), it would be also convenient to comment the importance of the quantification method.

- It would be desirable to show a graphical representation of FDG Time Activity Curves of test-retest studies, at least of one of the regions (representative). Likewise, it would be nice to show a representative test-retest PET images of one animal. All this information could be added as supplementary material.

Reviewer #3: Dear authors,

your study is very interesting and carried out very well. Nowadays it is very important to evaluate 18F-FDG PET images, from preclinical with animals to human studies, not just from a qualitative point of view. Some of the common quantitative parameters are still not robust enough to evaluate properly the true effects on the metabolism over time, for example when studying a new drug.

In this article the authors analyze the test-retest variance of many quantitative PET parameters (SUV, CMRglu, SUVxGly, SUV ratio on the pons or on the cerebellum) extracted from brain PET images performed twice on a small group of nonhuman primates (macaques) with a very controlled and reproducible protocol. They validated the results on an independent control group, too.

I think that the article is properly written down with a good depiction of the methods utilized and so, has to be accepted.

Major comment:

- Material and method: you describe in the “animals and housing” paragraph one anesthesia protocol for all procedures, but then it is slightly different for MRI and PET.

- I think that you can add a picture of the PET brain images and a figure of the curve obtained to show the quality of the scan.

- From a qualitative point of view there were any substantial difference between the scans? please add this information

- The mean glycemia level of this group of animals is comparable with other study with the same species? Please add this information

Minor comment:

- Line 63 “by the” is repeated twice

6. PLOS authors have the option to publish the peer review history of their article (what does this mean?). If published, this will include your full peer review and any attached files.

Reviewer #1: No

Reviewer #2: No

Reviewer #3: No

---

## [Author Response · Author response to Decision Letter 0]

2 Sep 2020

Reviewer #1: We thank Reviewer#1 for this comment.

Reviewer #2: 

-The authors have carried out a test-retest study controlling all the experimental conditions and have analyzed the reproducibility of the results by the study of intra-subject and inter-subject variability. However, another important measure which can be calculated from test-retest data is the reliability of the measurements by means intraclass correlation coefficient (ICC). Why authors don’t have calculated this coefficient?

-> We initially calculated the intraclass correlation coefficient (ICC) which is indeed a widely used reliability index in test-retest, intrarater, and interrater reliability studies. However, some of the ICC values in our study were negative. Although the ICC value should theoretically range between 0 and 1 per se (as it represents the proportionality of the total variance), its estimation may be negative. In these cases, it has been suggested to reset the ICC to 0 but such cases are then uninterpretable (Chen, 2017, Human Brain Mapping, doi: 10.1002/hbm.23909). We therefore chose another option and used a different measure of reliability. We report the mean absolute difference between values from the first and the second scan divided by the mean of both values to reflect the reliability of the measurements.

- In “PET acquisition” section (page 9, line 130) authors should detail the times at which arterial blood samples were collected to obtain arterial input function.

-> This information has been added in the text. Please see page 9 line 131

- For each PET study, authors acquired a 60 min dynamic emission scan (Page 8, line 120). Which temporal sequence did the authors use to reconstruct the studies? Please, explain in “PET acquisition“ section.

->We provided the framing in the “PET acquisition” paragraph. Please see page 9 line 123

- Which statistical test was performed to analyze the homogeneity of experimental conditions (body weight and injected dose)? Please, refer in “Statistical Analyses” section.

-> The homogeneity of experimental conditions (body weight, temperature and injected dose at test and retest) were analyzed using Wilcoxon signed-rank test. We have inserted this information in the “Statistical analysis” paragraph. Please see page 10 line 170.

- The authors have used a segmentation pipeline to automatically delineate the anatomical regions of interest for each macaque. However, it is usual to spatially normalize brain images to a common stereotaxic space to apply a predetermined template of VOIs.

Probably, the method used to quantify the different regions also influences the variability observed in the study.

Although the authors discuss the influence of different types of anesthesia on the results of the study (page 19, lines 349-352), it would be also convenient to comment the importance of the quantification method.

-> We have established that a large part of the test/retest variability was due to global rather than local effects, as the normalized SUV (SUVR) and especially the SUVRcerebellum show low test/retest variability.

In addition, for each animal, the same MRI segmentation was used for the analysis of the test and retest PET scan. Therefore, we would not expect much variability to be introduced by the MRI segmentation. On the other hand, registration errors between the two PET sessions might indeed contribute to the test/retest regional variability, although we expect this effect to remain marginal compared to more global experimental factors affecting SUV and CMRGlu parameters.

We inserted a comment in the Discussion page 19 line 347.

- It would be desirable to show a graphical representation of FDG Time Activity Curves of test-retest studies, at least of one of the regions (representative). Likewise, it would be nice to show a representative test-retest PET images of one animal. All this information could be added as supplementary material.

-> We have added a supplementary Figure (Fig. S1) displaying a)- time activity curves of a representative animal in the thalamus and pons regions and b)- corresponding SUV-normalized PET images in test and retest condition

Reviewer #3: 

- Material and method: you describe in the “animals and housing” paragraph one anesthesia protocol for all procedures, but then it is slightly different for MRI and PET.

-> We thank the reviewer for this relevant comment and have clarified this concern in the Methods section. Indeed, in both MRI and PET sessions, anesthesia was induced using a mixture of ketamine and xylazine (10 and 1mg/kg, respectively) in each animal. This information is now clearly reported p7 line 86. 

- I think that you can add a picture of the PET brain images and a figure of the curve obtained to show the quality of the scan. 

-> According to both reviewers suggestion, SUV values are now corrected and we have added a supplementary Figure (Fig. S1) displaying a)- time activity curves of a representative animal in the thalamus and pons regions and b)- corresponding SUV-normalized PET images in test and retest condition

- From a qualitative point of view there were any substantial difference between the scans? please add this information

-> Visual inspection of the raw and SUV PET images between test and retest scans revealed qualitative difference (see Example in Fig. S1). Actually, this difference was global and not regional. There is indeed a global factor difference between scan and rescan SUV images that is greatly decreased in the normalized SUVr images (especially in SUVr-cerebellum images). 

We inserted this information in the Result section. Please see p 12 line 213. 

- The mean glycemia level of this group of animals is comparable with other study with the same species? Please add this information 

-> Similar glucose levels in cynomolgus macaques (about 5 years old) have previously been reported in few studies (most metabolic studies in primates use the SUV parameter): Wu et al. (2012) reported a glycemia range of 2 to 4 mmol/L in male primates and Guo et al. (2014) found glycemia values around 4.7 mmol/L for instance.

We inserted this information in the Discussion section. Please see p 18 lines 324

Minor comment:

- Line 63 “by the” is repeated twice

-> Thank you, the typo has been corrected.

---

## [Editor Report · Decision Letter 1]

23 Sep 2020

Comparative test-retest variability of outcome parameters derived from brain [18F]FDG PET studies in non-human primates

PONE-D-20-17186R1

Dear Dr. Lavisse,

We’re pleased to inform you that your manuscript has been judged scientifically suitable for publication and will be formally accepted for publication once it meets all outstanding technical requirements.

Kind regards,

Matteo Bauckneht

Academic Editor

PLOS ONE
---

## [Editor Report · Acceptance letter]

25 Sep 2020

PONE-D-20-17186R1 

Comparative test-retest variability of outcome parameters derived from brain [18F]FDG PET studies in non-human primates 

Dear Dr. Lavisse:

I'm pleased to inform you that your manuscript has been deemed suitable for publication in PLOS ONE. Congratulations! Your manuscript is now with our production department. 

Kind regards, 

on behalf of

Dr. Matteo Bauckneht 

Academic Editor

PLOS ONE